# Genetic and pharmacological inhibition of microRNA-92a maintains podocyte cell cycle quiescence and limits crescentic glomerulonephritis

Carole Henique[1,2,3,4], Guillaume Bollée[1,2,5], Xavier Loyer[1,2], Florian Grahammer[6,7,8], Neeraj Dhaun[1,9], Marine Camus[1], Julien Vernerey[1], Léa Guyonnet[1,2], François Gaillard[1,2], Hélène Lazareth[1,2], Charlotte Meyer[8], Imane Bensaada[1,2], Luc Legrès[10], Takashi Satoh[11], Shizuo Akira[11], Patrick Bruneval[2,12,13], Stefanie Dimmeler[14], Alain Tedgui[1,2], Alexandre Karras[1,2,13,15], Eric Thervet[1,2,13,15], Dominique Nochy[2,12,13], Tobias B. Huber[6,7,8], Laurent Mesnard[16,17], Olivia Lenoir[1,2] & Pierre-Louis Tharaux [1,2,15]

Crescentic rapidly progressive glomerulonephritis (RPGN) represents the most aggressive form of acquired glomerular disease. While most therapeutic approaches involve potentially toxic immunosuppressive strategies, the pathophysiology remains incompletely understood. Podocytes are glomerular epithelial cells that are normally growth-arrested because of the expression of cyclin-dependent kinase (CDK) inhibitors. An exception is in RPGN where podocytes undergo a deregulation of their differentiated phenotype and proliferate. Here we demonstrate that microRNA-92a (miR-92a) is enriched in podocytes of patients and mice with RPGN. The CDK inhibitor p57$^{Kip2}$ is a major target of *miR-92a* that constitutively safeguards podocyte cell cycle quiescence. Podocyte-specific deletion of *miR-92a* in mice de-repressed the expression of p57$^{Kip2}$ and prevented glomerular injury in RPGN. Administration of an anti-miR-92a after disease initiation prevented albuminuria and kidney failure, indicating miR-92a inhibition as a potential therapeutic strategy for RPGN. We demonstrate that miRNA induction in epithelial cells can break glomerular tolerance to immune injury.

---

[1] Paris Cardiovascular Research Centre-PARCC, Institut National de la Santé et de la Recherche Médicale (INSERM), Paris 75015, France. [2] Paris Descartes University, Sorbonne Paris Cité, Paris 75006, France. [3] Institut Mondor de Recherche Biomédicale, team 21, Unité Mixte de Recherche (UMR) 955, Institut National de la Santé et de la Recherche Médicale (INSERM), Créteil 94000, France. [4] Université Paris-Est Créteil, Créteil 94000, France. [5] Centre de Recherche, Centre Hospitalier de l'Université de Montréal, Montréal, H2X 0A9 QC, Canada. [6] III. Medizinische Klinik, Universitätsklinikum Hamburg-Eppendorf, Hamburg 20246, Germany. [7] Department of Medicine IV, Medical Center–University of Freiburg, Faculty of Medicine, University of Freiburg, Freiburg im Breisgau, P.O. Box 79085, Germany. [8] BIOSS Centre for Biological Signalling Studies and Center for Biological Systems Analysis (ZBSA), Albert-Ludwigs-University, Freiburg 79104, Germany. [9] British Heart Foundation Centre of Research Excellence (BHF CoRE), Edinburgh, EH16 4TJ, UK. [10] Unité Mixte de Recherche (UMR_S) 1165, Institut National de la Santé et de la Recherche Médicale (INSERM), Plateforme MicroLaser Biotech, Paris 75010, France. [11] Laboratory of Host Defense, WPI Immunology Frontier Research Center (IFReC), Osaka University, Osaka 565-0871, Japan. [12] Department of Pathology, Hôpital Européen Georges Pompidou, Assistance Publique–Hôpitaux de Paris, Paris 75015, France. [13] Département Hospitalo-Universitaire, Paris Descartes University–Hôpitaux Universitaires Paris Ouest, Paris 75015, France. [14] Institute of Cardiovascular Regeneration, Centre for Molecular Medicine, Goethe University Frankfurt, Frankfurt 60590, Germany. [15] Renal Division, Hôpital Européen Georges Pompidou, Assistance Publique–Hôpitaux de Paris, Paris 75015, France. [16] Unité Mixte de Recherche (UMR) 702, Institut National de la Santé et de la Recherche Médicale (INSERM), Paris 75020, France. [17] Faculty of Medicine, University Pierre and Marie Curie, Paris 75020, France. Correspondence and requests for materials should be addressed to C.H. (email: carole.henique@inserm.fr) or to P.-L.T. (email: pierre-louis.tharaux@inserm.fr)

Necrotizing and crescentic rapidly progressive glomerulonephritis (RPGN) is one of the severest forms of glomerular disease. RPGN can occur in the setting of a number of immunological disorders including anti-glomerular basement membrane (anti-GBM) disease, anti-neutrophil cytoplasmic antibody (ANCA)-associated vasculitis, and systemic lupus erythematosus[1, 2]. Importantly, crescent formation within the glomerulus appears to occur downstream of the original inflammatory insult with common mechanisms that are independent of the initial injury. This might be because the pathogenesis of RPGN involves local factors as well as inflammatory cells and immune mediators.

The proliferation of podocytes[3, 4] and of parietal epithelial cells[5] plays a key role in extracapillary crescent formation. Mature

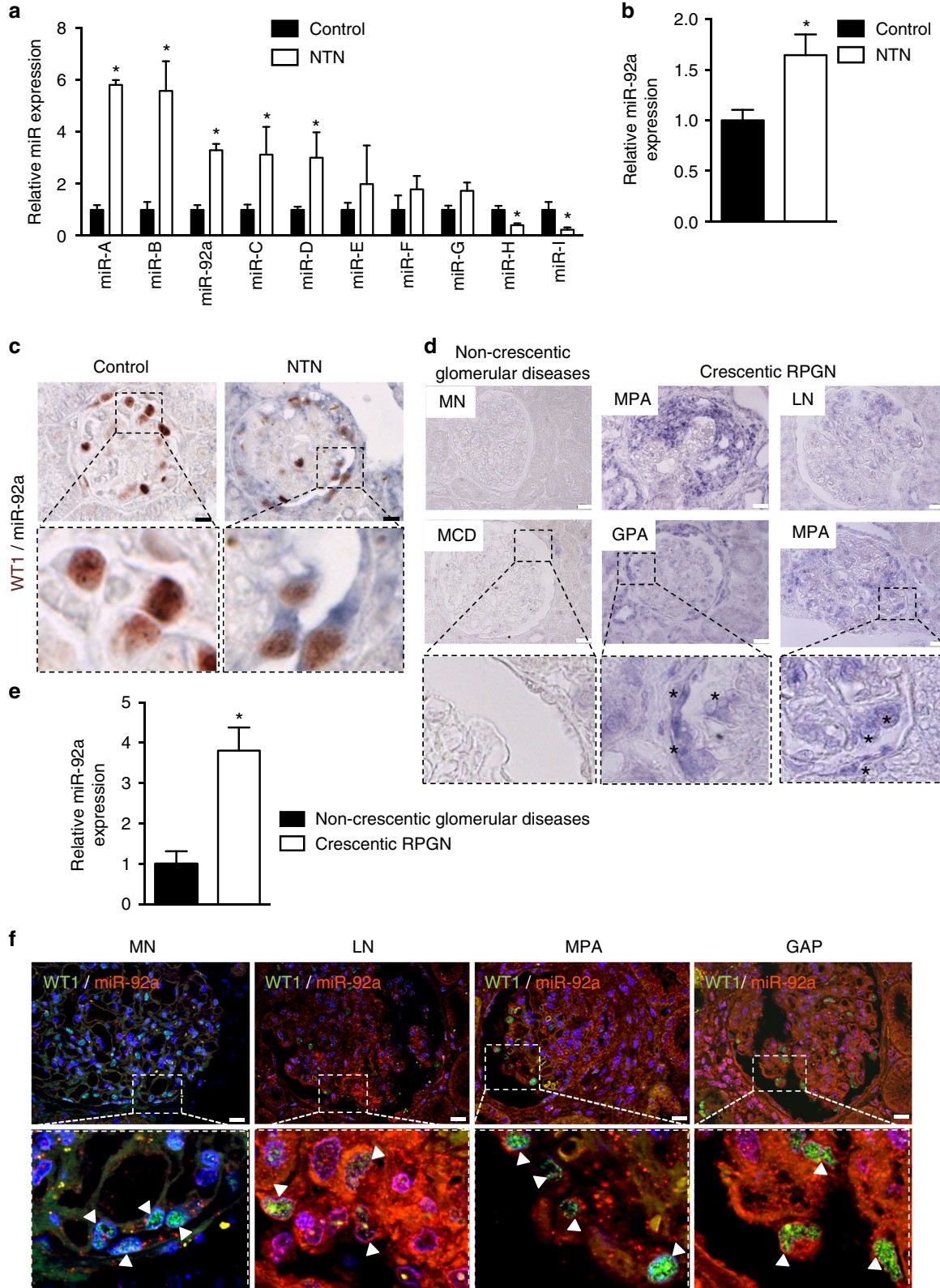

podocytes are glomerular epithelial cells that are normally growth-arrested because of the expression of cyclin-dependent kinase (CDK) inhibitors[6–10]. Under pathological conditions, podocytes may undergo mitosis but fail to complete cell division. Crescentic RPGN is an exceptional condition where podocytes undergo a dysregulation of their differentiated phenotype and start to proliferate and migrate resulting in the so-called "extracapillary glomerulopathy".

Studies in human kidney biopsies[3] and in a relevant mouse model[11] demonstrate that podocytes are dysregulated in RPGN; they lose their typical cell markers and switch to a proliferative phenotype. Convincing evidence for podocyte involvement in RPGN has come from two studies, one in which podocyte-specific deletion of the *Vhl* gene resulted in podocyte proliferation, crescent formation, and the rapid onset of renal failure[12], and the other in which podocyte-specific deletion of the *Egfr* gene prevented such features[13]. Numerous signaling pathways and proteins can be activated downstream of EGFR activation, including proteins of the signal transducer and activator of transcription (STAT) family, namely STAT5[14] and STAT3[15]. The STAT3-SH2 domain can directly bind phosphorylated EGFR at tyrosine 1068 and tyrosine 1086[16]. STAT3 transduces signals from growth factors and cytokines and plays an important role in development, cell growth, prevention of apoptosis, proliferation, and inflammation[17].

MicroRNAs (miRNAs) are endogenous, short-length, single-stranded non-coding RNAs that can disrupt gene expression by inducing translation inhibition and mRNA degradation. Recent evidence indicates that miRNAs may have a pivotal role in a number of renal disorders[18]. MiRNA profiling in isolated glomeruli from mice with nephrotoxic serum-induced crescentic nephritis (NTN) and control mice unraveled upregulation of miR92a. MiR92a belongs to the first discovered microRNA cluster as a potential human oncogene[19]. The miR-92a family is related to the formation of vascular endothelial cells[20, 21]. Aberrant expression of miR-92a family was detected in multiple cancers, and the disturbance of miR-92a family was related with tumorigenesis and tumor development[22].

Here we report that the abundance of miR-92a is high in kidney biopsies from patients diagnosed with crescentic RPGN, especially within podocytes, as well as in mice exposed to NTN. We next investigated the role of miR-92a in primary cultures of podocytes as well as in the NTN model of crescentic RPGN. Given the promoter region of the miR-17/92 gene contains a highly conserved functional STAT3-binding site[23], we examined miR-92a expression dependency on STAT3 in podocytes and in the context of severe extracapillary glomerulonephritis. We have discovered that the STAT3–miR-92a activation governs a dedifferentiation program in podocytes with acquisition of proliferative capability. We went on to examine the involvement miR-92a activation in the glomerular injurious process. Conditional podocyte-specific miR-92a deletion reduces albuminuria and glomerular injury and fully prevents renal failure after NTN. Inhibition of miR-92a in vitro upregulates the expression of its direct target the CDK inhibitor p57$^{Kip2}$ that regulates the podocyte cell cycle, and results in impairment of cell proliferation. To translate our findings to a potentially novel therapy for RPGN, we find that specific blockade of miR-92a in vivo markedly prevents albuminuria, crescent formation, and renal failure even when this strategy is initiated in a therapeutic manner after the onset of NTN.

## Results

**Glomerular miR-92a induction in mouse nephrotoxic nephritis.** We carried out miRNA profiling in isolated decapsulated glomeruli from NTS-challenged and control mice. We detected several significantly and differentially expressed miRNAs on day 10, a time when the onset of crescent formation is detectable with proliferating (Ki67+) podocytes (Fig. 1a). We then refined our analysis to select miRNA species that were enriched in podocytes using freshly sorted podocytes from nephritic and control mice with podocyte-specific GFP expression (*NPHS2*-Cre x mT/mG). MiR-92a was selectively induced in glomerular epithelial cells in nephritic mice as demonstrated both by in situ hybridization in WT1-expressing cells in kidney sections (Fig. 1b) and by RT-qPCR in freshly sorted podocytes from *NPHS2*-Cre x mT/mG mice (Fig. 1c).

**High expression of miR-92a in human kidneys with RPGN.** To evaluate the clinical relevance of our findings, we analyzed miR-92a expression in kidney biopsies from patients with RPGN and control patients with non-proliferative glomerulopathies (Table 1). Whereas miR-92a labeling was weak and restricted to the endothelium in control human kidneys (non-crescentic glomerular diseases) (Fig. 1d), miR-92a was significantly upregulated in kidney biopsies from patients with RPGN, regardless of etiology. In situ hybridization revealed the expression of miR-92a in glomerular epithelial cells of patients with RPGN, particularly in podocytes and crescents, and to a lesser extent in parietal epithelial cells (Fig. 1d). These results were independently confirmed by RT-qPCR analysis in laser capture microdissected glomeruli showing a 3.5- to 4-fold increase of miR-92a to U6snRNA ratio in RPGN cases compared to cases diagnosed with non-proliferative glomerular diseases (Fig. 1e).

This prominent induction of miR-92a was further demonstrated in WT1-expressing podocytes in kidney biopsies from individuals diagnosed with crescentic RPGN but not in those patients with non-proliferative glomerulopathies such as membranous nephropathy (Fig. 1f). In situ hybridization of U6snRNA

**Fig. 1** Increased miR-92a expression in crescents and podocytes during nephrotoxic nephritis and human crescentic glomerulonephritis. **a** Representative microRNA profiling on dynabeads-isolated glomeruli from control or NTS-challenged mice (NTN). Values are means ± s.e.m., *$p < 0.05$ vs. control condition ($n = 3$ samples per condition). **b** Relative abundance of miR-92a assessed by RT-qPCR in sorted podocytes from normal healthy mice (control) and NTS-challenged mice (NTN). Values are means ± s.e.m., *$p < 0.05$ vs. control condition ($n = 5$ mice per condition). **c** Double miR-92a in situ hybridization (blue staining) and WT1 staining (brown staining) of kidney sections from normal mice (control) and NTS-challenged mice (NTN). Pictures in the bottom show a higher magnification of the top panel. Scale bars, 10 μm. **d** miR-92a in situ hybridization of kidney sections from random biopsies from individuals diagnosed with non-crescentic glomerulopathies, including minimal change disease (MCD) and membranous nephropathy (MN), and from patients with RPGN of various etiologies including stage III and IV lupus nephritis (LN), microscopic polyangiitis (MPA), and granulomatosis with polyangiitis (GPA). Scale bars, 50 μm. The lower panel shows higher magnification of middle panels (black box). Black stars (*) show miR-92a-positive cells. **e** RT-qPCR analysis of the relative abundance of miR-92a in microdissected glomeruli from four patients with non-crescentic glomerulopathies (black bars) and six patients with crescentic RPGN (white bars). Values are means ± s.e.m., *$p < 0.05$ vs. non-crescentic glomerular diseases. **f** Fluorescent in situ hybridization of miR-92a (red) and WT1 (green) on patients biopsies described in **d**. Bottom panel shows higher magnification of top panel (white box). White arrows show colocalization of WT1-positive cells and miR-92a expression. Scale bars, 50 μm. Statistical analysis: Mann–Whitney test to compare two groups

**Table 1 Patients clinical details**

| Gender | Age | Diagnosis | Group of patients | First episode/ relapse | Treatment | Urinary protein (g/mmol creatinine) | eGFR (ml/min/1.73 m²) |
|---|---|---|---|---|---|---|---|
| F | 29 | MCD | Control | Relapse | Corticosteroids | 1.07 | 79 |
| F | 30 | MCD | Control | Relapse | No treatment | 0.19 | 73 |
| M | 66 | MCD | Control | First episode | No treatment | 0.56 | 55 |
| F | 31 | MCD | Control | First episode | No treatment | NA | 94 |
| F | 30 | MN | Control | First episode | No treatment | 0.08 | 89 |
| M | 46 | MN | Control | First episode | No treatment | 0.6 | 90 |
| M | 50 | MN | Control | First episode | No treatment | >3 | 72 |
| M | 62 | MPA MPO-ANCA-positive | RPGN | First episode | No treatment | 0.1 | 63 |
| M | 78 | MPA PR3-ANCA-positive | RPGN | First episode | Corticosteroids | 0.23 | 39 |
| M | 52 | MPA MPO-ANCA-positive | RPGN | First episode | No treatment | 0.41 | 47 |
| F | 46 | MPA MPO-ANCA-positive | RPGN | First episode | No treatment | 0.26 | 18 |
| M | 47 | MPA MPO-ANCA-positive | RPGN | First episode | No treatment | 0.27 | 6 |
| F | 58 | MPA MPO-ANCA-positive | RPGN | First episode | No treatment | Traces | 6 |
| M | 44 | MPA MPO-ANCA-positive | RPGN | First episode | No treatment | 0.08 | 55 |
| M | 68 | GPA PR3-ANCA-positive | RPGN | First episode | Corticosteroids | 0.15 | 58 |
| M | 71 | GPA PR3-ANCA-positive | RPGN | First episode | No treatment | 0.56 | 75 |
| M | 55 | GPA PR3-ANCA-positive | RPGN | Relapse | Corticosteroids | 5 | 52 |
| F | 35 | LN (class IV) | RPGN | Relapse | Corticosteroids | 0.34 | 35 |
| F | 27 | LN (class IV) | RPGN | Relapse | Corticosteroids | 0.26 | 41 |
| M | 17 | LN (class IV) | RPGN | First episode | No treatment | 0.13 | 37 |
| F | 41 | LN (class III) | RPGN | Relapse | Corticosteroids | 0.19 | 25 |
| F | 25 | LN (class III) | RPGN | First episode | No treatment | 0.25 | 50 |

Patients' characteristics at the time of the kidney biopsy used in the study. The eGFR is calculated according to the Chronic Kidney Disease Epidemiology Collaboration (CKD-EPI) equation
MCD minimal change disease; MN membranous nephropathy; MPA microscopic polyangiitis; GPA granulomatosis with polyangiitis (GPA); ANCA anti-neutrophil cytoplasmic antibodies; MPO myeloperoxidase; PR3 proteinase 3; LN class III and class IV lupus nephritis, control for non-crescentic glomerular disease and RPGN for crescentic RPGN

was used as a control (Supplementary Fig. 1a). Moreover, utilizing RT-qPCR analysis of miRNA in whole kidney biopsy sections (a more practical alternative to laser capture micro-dissection of glomeruli or in situ hybridization), we showed that the expression of miR-92a was 2- to 7.5-fold higher in samples from patients with RPGN than in those from patients with other chronic proteinuric glomerular diseases (minimal change disease (MCD) and membranous nephropathy (MN)) (Supplementary Fig. 1b) with no overlap of relative miR-92a levels between the RPGN and non-RPGN groups. Interestingly, a similar pattern of miR-92a expression was observed in all kidney samples from patients with RPGN regardless of etiology.

As miR-92a is part of the miR-17–92 cluster that encodes a polycistronic transcript that produces six individual mature miRNAs[24], we expected increased glomerular expression of all members of the cluster. Surprisingly, miR-92a was the only member to be dysregulated in human (Supplementary Fig. 1c) and murine RPGN (Supplementary Fig. 1d).

These studies show that miR-92a is highly abundant in conditions associated with glomerular epithelial cell proliferation and crescent formation.

**miR-92a expression is modulated downstream of the STAT3 cascade in podocytes**. We focused our analysis on STAT3-dependent miRNAs given our evidence for EGFR and STAT3 activation in freshly isolated glomeruli of NTS-challenged and control mice. Phosphorylation of EGFR at tyrosine 1068, a marker of EGFR activation, and phosphorylation of STAT3 at tyrosine 705, a marker of STAT3 activation, were stimulated by NTS administration (Supplementary Fig. 2a, b). Furthermore, nephritic mice displayed a 10-fold increase in podocyte nuclear localization of phosphorylated STAT3 (pY705) compared to healthy mice (Supplementary Fig. 2c, d). Notably, STAT3 activation was recently shown to aggravate experimental crescentic glomerulonephritis[25] and is also known to activate several

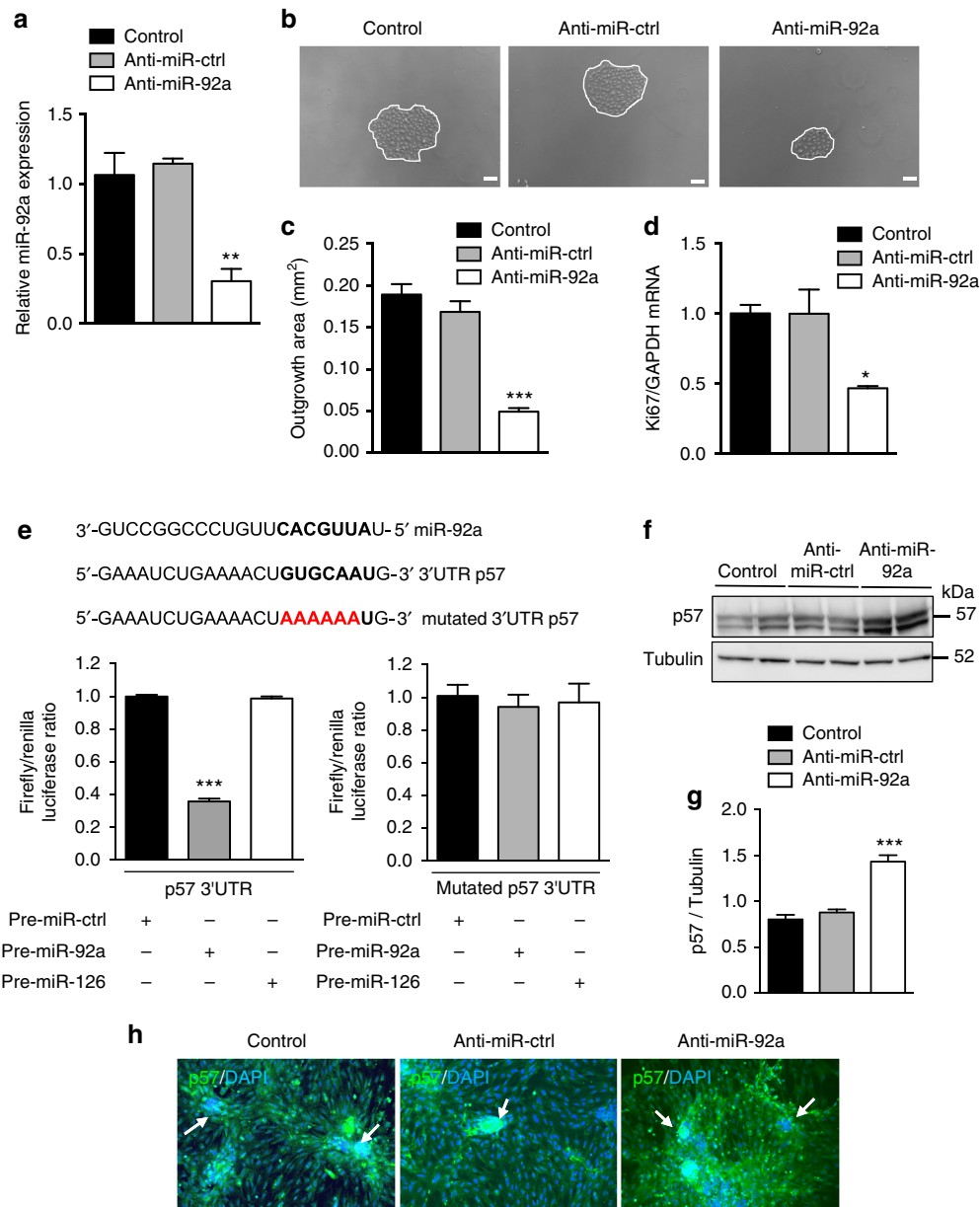

**Fig. 2** Inhibition of miR-92a in podocytes upregulates its target p57 and impairs proliferation. **a** RT-qPCR analysis of the relative abundance of miR-92a in podocytes transfected with an anti-miR-control (anti-miR-ctrl) or an anti-miR-92a. Values are normalized to U6snRNA and are relative to control (non-transfected cells). **b, c** Representative pictures (**b**) and quantification (**c**) of podocyte proliferation assay involving decapsulated mouse glomeruli. Podocyte proliferation was assessed after 4 days. Scale bars, 50 μm. **d** RT-PCR analysis of Ki67 mRNA abundance in intact and anti-miR-ctrl primary podocytes or in anti-miR-92a primary podocytes. **e** Dual luciferase assay wild-type or mutated p57 3′UTR in HEK293 cells transfected with pre-miR-ctrl, pre-miR-92a, or pre-miR-126. $n = 2$ independent experiments (each experiment assayed each condition in triplicate). p57 3′UTR miR-92a-binding sequence binding site is indicated in bold and mutated sequence is labeled in red. ***$p < 0.001$ vs. 3′UTR +Pre-miR- ctrl. **f, g** Western blot analysis (**f**) and quantification of p57 protein abundance (**g**) in untreated or anti-miR-ctrl primary podocytes vs. in anti-miR-92a primary podocytes. Tubulin is shown as a loading control. **h** Staining of p57 protein (green) in podocyte outgrowths. DAPI-stained nuclei (blue). Adherent glomeruli are indicated by arrows. Scale bars, 100 μm. Statistical analysis: Kruskal–Wallis one-way analysis of variance followed by Dunn's multiple comparaison test. Values are means ± s.e.m. ($n = 4$ per group). *$p < 0.05$, **$p < 0.01$, and ***$p < 0.001$ vs. control podocytes (control)

miRNAs in a number of proliferative disorders[26–29]. We focused our bioinformatics analysis on those STAT3-dependent miRNAs whose predicted targets control cell proliferation (Supplementary Fig. 2e). MiR-92a was the most differentially upregulated candidate fulfilling these criteria. A bioinformatics screen for STAT3-binding sites in the promoter of the chromosome 13 open reading frame 25 (C13orf25) containing the miR-17–92 cluster revealed a highly conserved binding site. Furthermore, this binding site for STAT3 was previously functionally confirmed by Brock et al. with a reporter gene assay study. The authors showed a down-regulation of luciferase activity when the predicted STAT3-binding site on C13orf25 promoter was mutated (mutated promoter construct)[23].

We next investigated upstream activators of the STAT3-miR-92a cascade in primary cultures of podocytes. STAT3 signaling is activated by ligands binding to the gp130 receptor and EGFR[30–32]

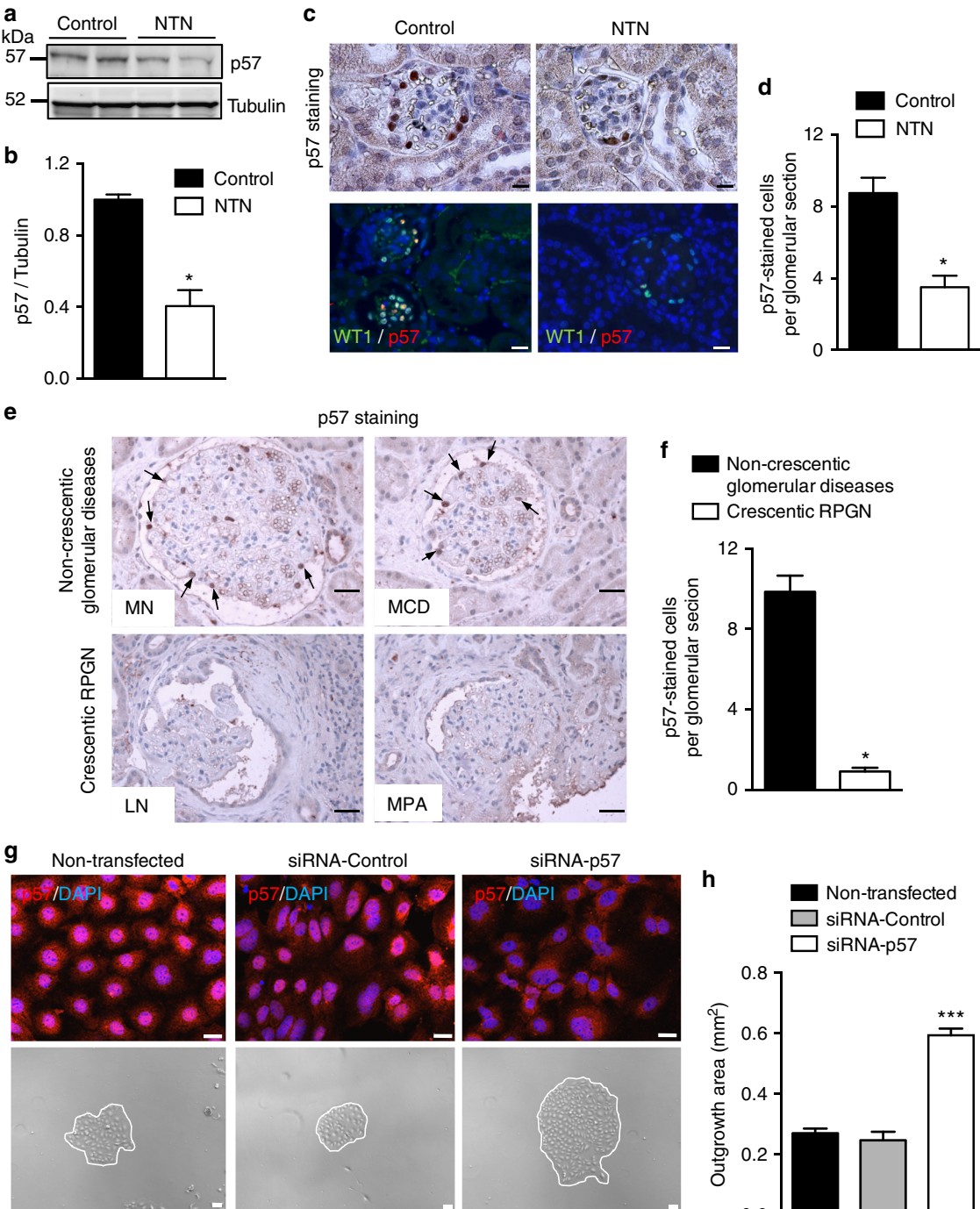

**Fig. 3** p57$^{Kip2}$ expression is lost during RPGN and p57$^{Kip2}$ silencing decreases podocyte proliferation. **a**, **b** Western blot analysis (**a**) and quantification (**b**) of p57 on magnetic beads isolated glomeruli from unchallenged (control) or NTS-challenged (NTN) mice. Values are means ± s.e.m. ($n = 5$ per group). *$p$ < 0.05 vs. control. **c** Immunostaining of p57 (strong brown staining) and immunofluorescent p57 (red)/WT1 (green) staining in kidney sections from normal mice (control) and NTS-challenged mice (NTN). Scale bars, 10 µm. **d** Quantification of p57-positive cells per glomerular section in mice described in **a**. **e** Representative photomicrographs of p57 immunostaining (strong brown staining) in kidney sections from random biopsies from individuals diagnosed with non-crescentic glomerulopathies, including minimal change disease (MCD) and membranous nephropathy (MN), and from patients with RPGN of various etiologies including stage III and IV lupus nephritis (LN) and microscopic polyangiitis (MPA). Scale bars, 50 µm. **f** Quantification of p57-positive cells per glomerular section in biopsies described in **e**. Values are means ± s.e.m. *$p$ < 0.05 vs. non-crescentic glomerular diseases. **g** Representative pictures of p57 staining (red) and podocyte proliferation assay in control cells (non-transfected) or transfected with a control siRNA (siRNA-control) or a siRNA for p57 (siRNA-p57). DAPI-stained nuclei (blue). Top panel (scale bars, 50 µm); bottom panel (scale bars, 100 µm). **h** Quantification of podocyte proliferation assay involving decapsulated mouse glomeruli. Podocyte proliferation was assessed after 4 days. Values are means ± s.e.m. ($n = 6$ per group). ***$p$ < 0.001 vs. non-transfected cells. Statistical analysis: Kruskal–Wallis one-way analysis of variance followed by Dunn's multiple comparaison test or Mann–Whitney test to compare groups. Values are means ± s.e.m

or IL-6[33–35]. As previously found, we measured activation of the *Hbegf* gene in cultured podocytes, as found in crescent[13]. We also measured a fourfold increase in *IL6* mRNA expression by primary podocytes undergoing a dedifferentiation and proliferative program (Supplementary Fig. 3a).

To examine whether IL-6 and EGFR activate STAT3 in podocytes, we blocked IL-6 and EGFR in cultured primary podocytes and performed western blotting to determine STAT3 phosphorylation (Tyr705), which is a marker of STAT3 activation. Cultured podocytes displayed autocrine constitutive

STAT3 (Supplementary Fig. 3b, c, d, f) and EGFR activation (Supplementary Fig. 3d, e). We found that both addition of exogenous recombinant IL-6 or HB-EGF-stimulated STAT3 phosphorylation and miR-92a expression (Supplementary Fig. 3g, h). Furhermore, an anti-mIL-6 monoclonal antibody and a specific inhibitor of EGFR kinase, AG1478 impaired STAT3 phosphorylation, miR-92a expression, and Ki67 levels (Supplementary Fig. 3f, h, i). These data indicate that the IL-6 receptor (IL-6R) and EGFR pathways are tonically activated by autocrine ligands synthesized by activated podocytes. Taken together, these

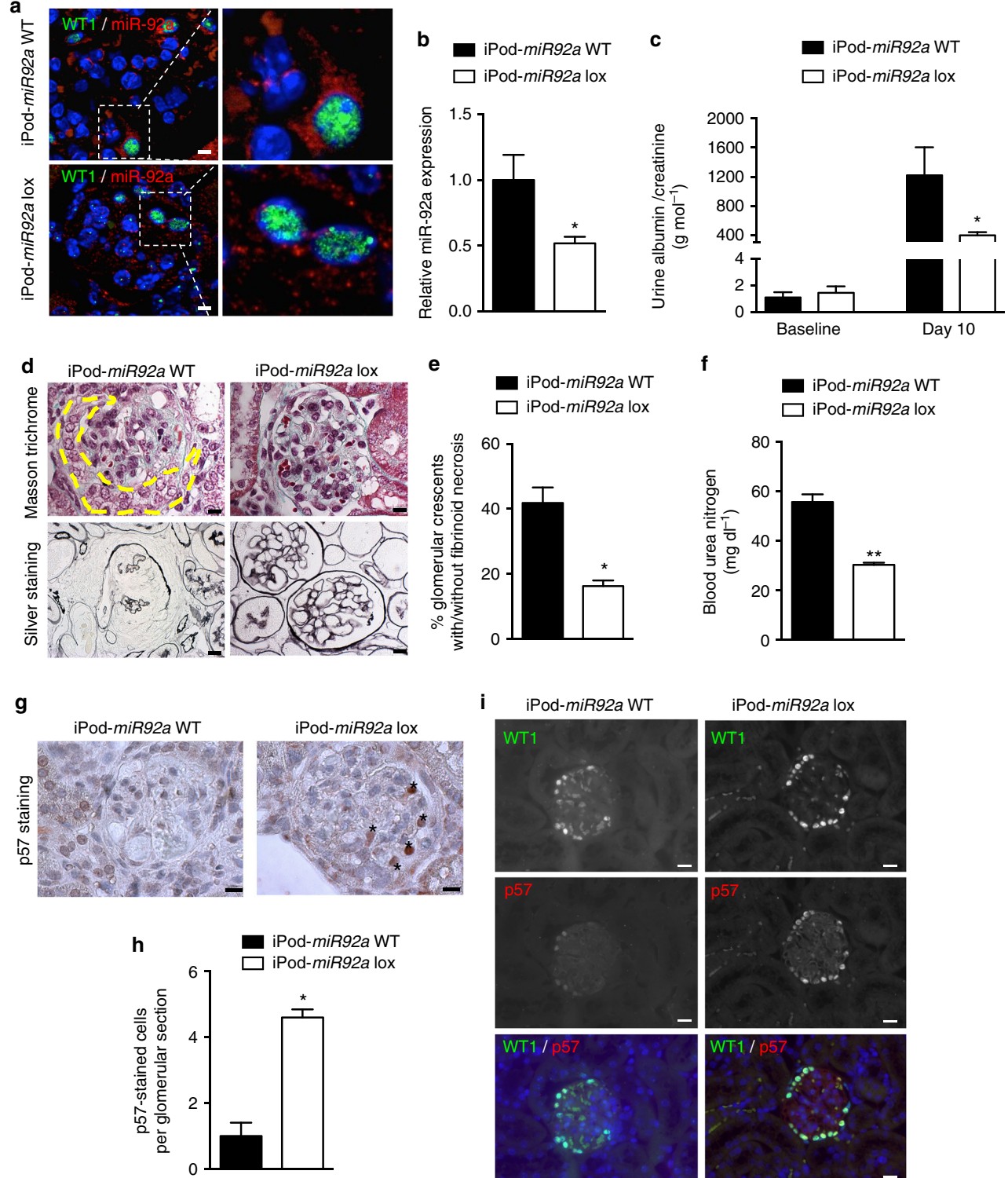

findings suggest that both EGFR and IL-6R stimulate the STAT3-miR-92a cascade in activated proliferating podocytes.

To further confirm the mechanisms whereby miR-92a is induced in vivo, we went on to abolish STAT3 expression specifically in podocytes by generating mice with floxed *Stat3* and podocyte-specific expression of Cre (*Pod*). Double staining of STAT3 protein and podocalyxin (a cell surface marker of podocytes) on kidney sections revealed marked constitutive expression of STAT3 in podocytes of glomeruli from nephritic *Pod-Stat3 WT* mice but STAT3 staining was almost absent in podocytes from *Pod-Stat3 lox* mice (Supplementary Fig. 4a). Deletion of STAT3 was also confirmed in cultures of primary podocytes isolated from *Pod-Stat3 lox* or *Pod-Stat3 WT* mice for which the prurity of the primary culture was confimed by examining podocin, nephrin, and WT1 staining (Supplementary Fig. 4b); western blot analysis showed ~90% reduction in STAT3 expression in cultured podocytes from *Pod-Stat3 lox* compared to controls (Supplementary Fig. 4c, d).

Genetic *Stat3* allele deletion limited miR-92a expression in primary culture of podocytes (Supplementary Fig. 4e). A similar effect was observed in *Pod-Stat3 WT* cells treated with Stattic, a pharmacological STAT3 inhibitor (not shown)[36].

Given we had found STAT3 to be a strong modulator of miR-92a in cultured podocytes, we studied the expression of miR-17/92 in experimental NTN on *Pod-Stat3 lox* mice. MiR-92a upregulation was confirmed in vivo by miR-92a RT-qPCR in isolated glomeruli (Supplementary Fig. 4f) and by in situ hybridization (Supplementary Fig. 4f, g) that showed a widespread high abundance of miR-92a throughout glomerular sections from NTS-challenged nephritic Pod-*Stat3* WT mice, particularly in glomerular epithelial cells. MiR-92a glomerular expression was not induced in *Pod-Stat3 lox* mice despite NTS challenge. Overall, miR-92a expression was significantly lower in glomeruli isolated from *Pod-Stat3 lox* mice compared to that measured in *Pod-Stat3 WT* podocytes.

**miR-92a inhibition preserves p57^Kip2 expression and limits podocyte proliferation.** To decipher the role of miR-92a in podocyte function, we inhibited miR-92a expression in primary cultures of podocytes (Fig. 2a). We established an in vitro assay for podocyte crescent formation and measured outgrowth of podocytes from isolated, decapsulated mouse glomeruli[13]. Both podocyte outgrowth area and the abundance of Ki67 mRNA were lower in podocytes transfected with anti-miR-92a than in control cells transfected with an anti-miR control (Fig. 2b, c, d). Therefore, miR-92a is involved in the regulation of podocyte proliferation. We next searched for potential targets of miR-92a using the target prediction algorithms miRWalk and TargetScan[37, 38]. We focused on genes relevant to the maintenance of podocyte quiescence. Among potential candidates, the p57/Kip2/Cyclin-dependent kinase inhibitor 1C is a member of the Cip/Kip family and is a strong inhibitor of several G1 cyclin/Cdk complexes and a negative regulator of cell proliferation[39, 40]. Overexpression of p57^Kip2 leads to G1 phase cell cycle arrest. This protein has been shown to be constitutively expressed in mature podocytes[10, 41], an observation that we confirmed. Interestingly, impairment in p57^Kip2 expression during glomerular disease is associated with a high rate of podocyte proliferation[6]. p57^Kip2 displays a 3′UTR miR-92a-binding sequence. Utilizing a 3′UTR luciferase assay, we confirmed that miR-92a directly targets p57^Kip2 (Fig. 2e and Supplementary Fig. 6a, b, c). Furthermore, we showed by western blot and immunofluorescence that the abundance of p57^Kip2 was higher in anti-miR-92a-transfected podocytes and correlated with a lower rate of podocyte proliferation (Fig. 2f, g, h).

We next studied whether the p57^Kip2 expression pattern in podocytes during RPGN corresponded to the loss of expression observed in proliferating cultured podocytes with high miR-92a expression. We found significantly reduced p57^Kip2 abundance in glomerular lysates from NTS-challenged mice (Fig. 3a, b) and immunostaining for p57^Kip2 in kidney sections of mice confirmed this loss was exclusively from the podocytes (Fig. 3c, d). Importantly, in keeping with our pre-clinical findings, we found almost complete disappearance of p57^Kip2 expression in glomeruli from individuals diagnosed with crescentic RPGN due to ANCA vasculitis and lupus nephritis (Fig. 3e, f).

In order to investigate whether downregulation of p57^Kip2 is associated with defective cell cycle exit of podocytes, we knocked down p57^Kip2 in primary mouse podocytes cultures (Fig. 3g and Supplementary Fig. 6d, e) that significantly boosted podocyte proliferation (Fig. 3g, h). These data indicate that p57^Kip2 is a "master safeguard" of podocyte quiescence.

**miR-92a deletion prevents crescentic glomerulonephritis and renal failure.** To determine if miR-92a expression in podocytes in vivo is necessary for crescent formation and renal failure, we selectively deleted *miR-92a* from podocytes. We used a conditional expression model (Tet-On system) to achieve temporal podocyte-specific deletion of the *miR-92a* gene in mice. Mice of all genotypes were born at the expected Mendelian frequency and appeared healthy. Ten weeks after doxycycline administration, marked reduction of miR-92a abundance in podocytes was achieved (Fig. 4a), with most of residual signal being in endothelial cells. Overall, a ~50% reduction in relative miR-92a expression was achieved in glomeruli (Fig. 4b). Podocin-rtTA-Tet-O-Cre miR-92a loxP/loxP (iPod-miR92a lox) mice had normal kidney histology (Supplementary Fig. 7) and albuminuria that was within the physiological range (Fig. 4c). iPod-miR92a lox males and Podocin-rtTA-Tet-O-Cre miR-92a wt/wt (iPod-miR92a wt) gender-matched littermates were subjected to severe, life-threatening NTN with high-dose NTS. Podocyte-specific deletion of *miR-92a*-alleviated albuminuria (Fig. 4c), crescent formation (Fig. 4d, e), the rise in BUN (Fig. 4f), and significantly protected mice from p57^Kip2 loss in podocytes (Fig. 4g–i).

**Fig. 4** miR-92a-specific deletion in podocytes reduces nephrotoxic nephritis. **a** Fluorescent in situ hybridization of miR-92a (red) and WT1 (green) on kidney sections from NTS-challenged iPod-miR92a WT mice and iPod-*miR92a* lox mice. DAPI-stained nuclei (blue). Right panel shows higher magnification of the left panel (white box). Scale bars, 50 µm. **b** RT-qPCR analysis of the relative abundance of *miR-92a* in freshly isolated glomeruli from NTS-challenged iPod-*miR92a* WT mice and NTS-challenged iPod-*miR92a* lox mice. **c** Urinary albumin excretion rates at baseline and 10 days after NTS injection. **d** Masson trichrome- and silver-stained kidney sections of glomeruli from NTS-challenged iPod-*miR92a* WT mice and iPod-*miR92a* lox mice at day 10 after NTS injection. Scale bars, 10 µm. **e** Proportion of crescentic glomeruli in NTS-challenged iPod-*miR92a* WT and iPod-*miR92a* lox mice at day 10 after NTS injection. **f** Blood urea nitrogen concentration at day 10 after NTS injection in iPod-*miR92a* WT and iPod-*miR92a* lox mice. **g** Immunostaining of p57 (strong brown staining, *) in kidney sections from mice described in **a**. Scale bars 10 µm. **h** Quantification of p57-positive cells per glomerular section in mice described in **a**. **i** Representative photomicrophotographs of dual immunofluorescent sating of p57 (red) and WT1 (green) in kidney sections from mice described in **a**. Statistical analysis: Mann–Whitney test to compare two groups. Values are means ± s.e.m. (n = 7 per group). *p < 0.05, **p < 0.01 vs. iPod-*miR92a* WT mice

**Anti-miR92a protects from crescentic glomerulonephritis and kidney failure**. Our data suggest that RPGN is associated with increased miR-92a expression and activity in podocytes and that this may be pathogenic in disease progression. Hence, we went on to test whether pharmacological inhibition of the miR-92a–p57$^{Kip2}$ pathway protected from renal injury by using

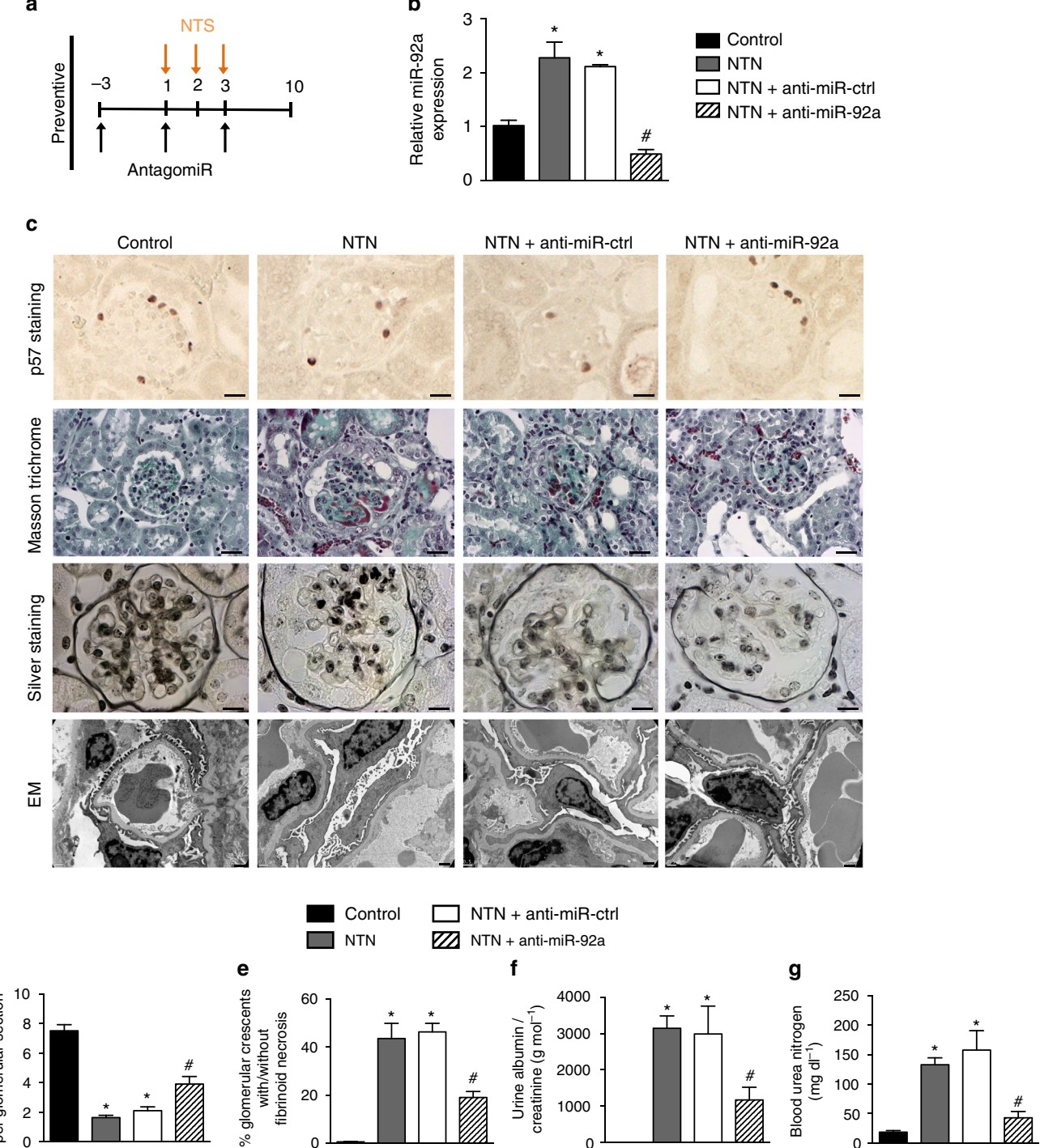

**Fig. 5** Silencing miR-92a prevents kidney injury in a mouse model of nephrotoxic nephritis. **a** Study design of the in vivo preventative antagomir experiment. **b** RT-qPCR analysis of the relative abundance of miR-92ain freshly isolated glomeruli from normal mice (control), NTS-challenged mice (NTN), NTS-challenged mice treated with anti-miR-control (NTN + anti-miR-ctrl) and NTS-challenged mice treated with anti-miR-92a (NTN + anti-miR-92a) after 10 days. All values are normalized to U6snRNA and are relative to control. Values are means ± s.e.m. ($n = 4$ per group). *$p < 0.05$ vs. control, #$p < 0.05$ vs. NTN alone. **c** Representative photomicrographs of p57 staining (scale bars, 10 μm), Masson trichrome- (scale bars, 20 μm), silver-stained kidney sections (scale bars, 10 μm) and transmission electron microscopy (scale bars, 0.5 μm) from groups of mice described in **a**. **d** Quantification of p57$^{Kip2}$-positive cells per glomerular section in mice described in **a**. **e** Proportion of glomerular crescents in kidney sections, **f** albuminuria, and **g** blood urea nitrogen concentrations in mice as described in **a**. Values are means ± s.e.m. ($n = 4$ per group). *$p < 0.05$ vs. control, #$p < 0.05$ vs. NTN alone. Statistical analysis: Kruskal–Wallis one-way analysis of variance followed by Dunn's multiple comparaison test

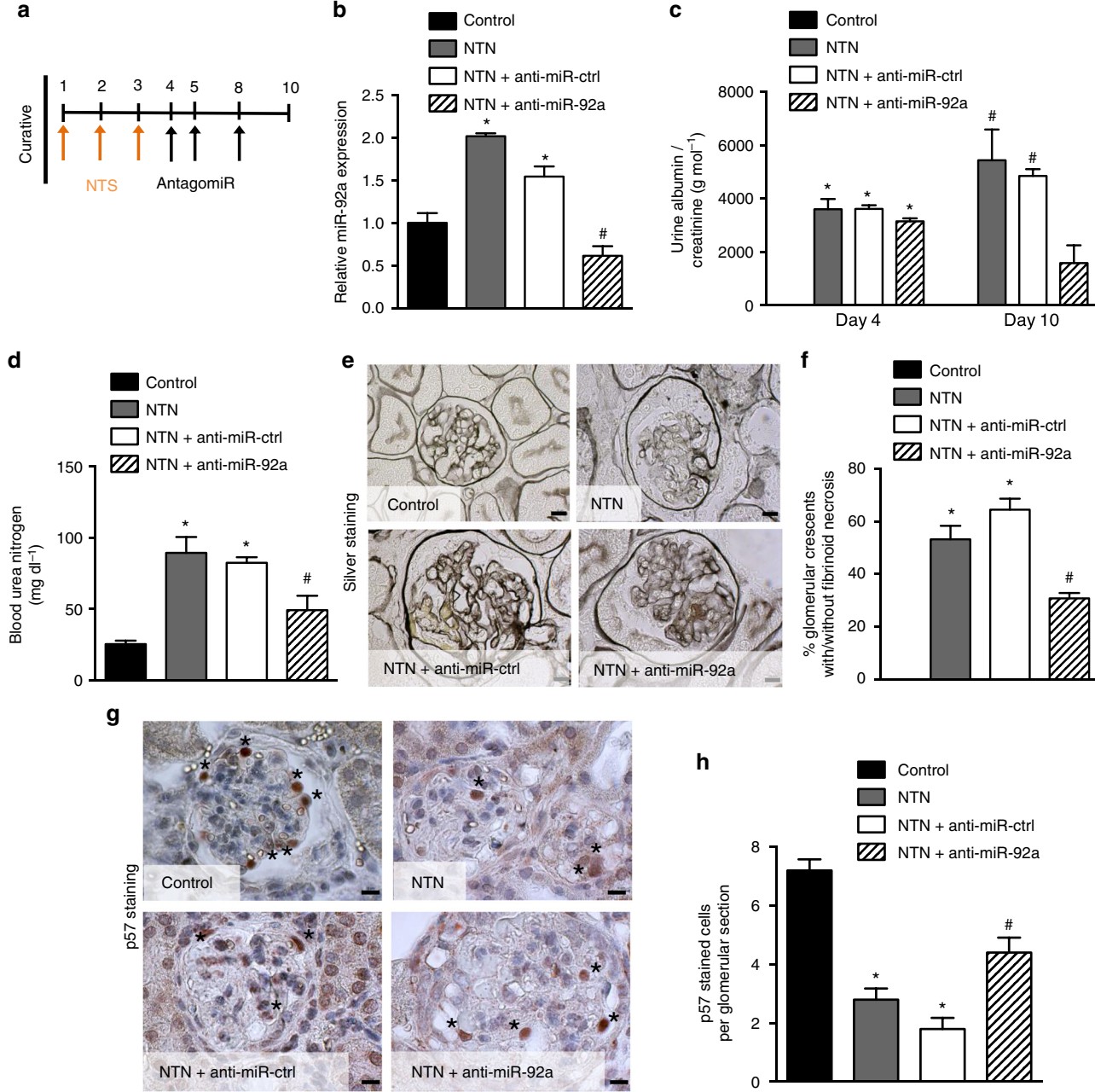

**Fig. 6** miR-92a in vivo silencing abolishes nephrotoxic nephritis development. **a** Study design of the in vivo antagomir experiment. **b** Relative miR-92a expression in dynabeads-isolated glomeruli from normal mice (control), NTS-challenged mice (NTN), NTS-challenged mice treated with anti-miR-control (NTN + anti-miR-ctrl), and NTS-challenged mice treated with anti-miR-92a (NTN + anti-miR-92a) after 10 days. All values are normalized to U6 and are relative to control. Values are means ± s.e.m. ($n = 5$ per group). *$p < 0.05$ vs. control, #$p < 0.05$ vs. NTS alone. **c** Urinary albumin excretion rates at day 4 (before antagomir injection) and at day 10 after NTS injection. **d** Blood urea nitrogen concentration at day 10 after NTS injection in control or NTS-challenged mice. **e** Silver-stained kidney sections of mice described in **b**. Scale bars, 10 μm. **f** Proportion of crescentic glomeruli in kidney from mice described in **b**. Values are means ± s.e.m. ($n = 10$ mice per group). **g** Representative immunostaining of p57 (strong brown staining) in kidney sections from mice described in **a**. Scale bars, 10 μm. **h** Quantification of p57-positive cells per glomerular section in mice described in **a**. Values are means ± s.e.m. ($n = 10$ per group). *$p < 0.05$, #$p < 0.05$ vs. NTS alone (NTN). Statistical analysis: Kruskal–Wallis one-way analysis of variance followed by Dunn's multiple comparaison test

chemically engineered oligonucleotides, termed "antagomirs" (anti-miRs), efficient and specific silencers of endogenous miR-NAs in mice. Anti-miRs are taken up into the kidney cortex after intravenous injection into mice[42].

We first administered an anti-miR-92a strategy in a preventative fashion. Anti-miR-92a injections to mice inhibited miR-92a expression in glomeruli during NTN (Fig. 5a, b) without modifying the expression of other miRNAs from the 17–92

cluster (not shown). Induction of NTN was associated with a low abundance of p57[Kip2] in podocytes that was partially but significantly rescued by delivery of anti-miR-92a (Fig. 5c, d and Supplementary Fig. 8a). Importantly, NTS-induced glomerular injury was less severe in mice that received anti-miR92a (NTN +anti-miR92a) than in mice that received a control anti-miR (NTN+anti-miR-ctrl) or in mice that received vehicle alone (NTN). Indeed, the proportion of glomerular crescents was ~70%

lower in NTN+anti-miR92a mice than in either control condition (Fig. 5c, e). This improvement in kidney structure was also reflected by less severe damage to the podocyte ultrastructure in anti-miR-92a-injected mice relative to untreated and anti-miR-ctrl injected littermate controls (Fig. 5c). These improved structural parameters corresponded to improved functional parameters: urinary albumin excretion was lower (Fig. 5f), and kidney dysfunction minimal (Fig. 5g) in anti-miR-92a-treated mice.

We next administered anti-miR-92a, on day 4 after infusion of NTS (Fig. 6a). This time-point was chosen as it is clinically relevant, associated with nephrotic range albuminuria and high serum creatinine and BUN. As previous, this regimen was compared with the effects of vehicle alone and with the administration of an anti-miR-ctrl. Anti-miR-92a given after NTS still effectively inhibited glomerular miR-92a levels at day 10 (Fig. 6b) and these mice displayed a marked reduction in albuminuria (Fig. 6c). Furthermore, whereas vehicle-only-treated mice (NTN) or anti-miR-ctrl-treated mice (NTN+anti-miR-ctrl) developed rapid and life-threatening kidney failure, mice treated therapeutically with anti-miR-92a had BUN levels within the normal range (Fig. 6d). This functional protection conferred by anti-miR-92a administration was associated with marked alleviation of histologic damage as measured using silver staining of renal cortex (Fig. 6e, f) and significantly preserved p57[Kip2] expression in podocytes (Fig. 6g, h and Supplementary Fig. 8b).

## Discussion

RPGN with extracapillary proliferation of epithelial glomerular cells is an area of unmet clinical need. Despite current immunosuppressive therapies, a significant number of patients with RPGN fail to respond to treatment and their disease progresses to end-stage kidney disease with its associated significant morbidity and mortality. RPGN involves the response of podocytes to immune injury. Therefore, better understanding of the mechanisms that regulate podocyte function are critical. Here we explored the role of miRNAs. Furthermore, the mechanisms whereby terminally differentiated, post-mitotic podocytes reenter cell cycle upon immune-mediated stress are unknown.

Our data demonstrate the involvement of miR-92a in the deleterious response to immune injury that leads to glomerular destruction and functional demise. Our investigation further identified a key miR-92a target, the p57/Kip2/Cyclin-dependent kinase inhibitor 1C, which is involved in cell cycle regulation and control of the quiescent state of podocytes.

MiR-92a is part of the miR-17–92 cluster, which contains six miRNAs[24]. The human MIR17(MIR17HG) cluster has been linked to developmental, apoptotic, and oncogenic pathways in other organs, and the locus is conserved between mouse and man. Mice deficient for miR-17/92 die shortly after birth and have cardiac and lung abnormalities. Thus, this cluster plays an important role during development. Among the members in the cluster, miR-92a is the least characterized. Whereas endothelial miR-92a has been characterized with pro-inflammatory and anti-angiogenic effects[43–45], ours is the first description of epithelial induction in post-mitotic cells. As miR-92a is part of the miR-17–92 cluster that encodes a polycistronic transcript that produces six individual mature miRNAs[24], we expected increased glomerular expression of all members of the cluster. Surprisingly, miR-92a was the only member to be dysregulated in human and murine RPGN. In fact, this phenomenon was also previously found in experimental atherosclerosis[45]. Moreover, these in vivo data are consistent with the recent demonstration that expression of individual miRNAs from pri-miR-17/92 is dynamically regulated[46].

MiR-92a was significantly overexpressed in human glomeruli from patients with RPGN compared to those with other non-proliferative glomerular diseases. This difference highlights the difference in pathogenesis of these diseases, the former being critically dependent on loss of podocyte post-mitotic quiescence. However, miR-92a expression level did not correlate with histological variant of crescent (cellular vs. fibrocellular), proteinuria, or kidney function. This may be explained by clinical heterogeneity in terms of stage of disease at the time of diagnosis or relationship between miRNA expression and cell density. For example, we observed intense miR92a expression in fully constitued crescents and less in pseudo-crescents (when a paucity of cells are proliferating) and fibrocellular cellular crescents (when many epithelial cells have already disappeared).

We identified a miR-92a target relevant to podocyte proliferation. In contrast to immature podocytes, which proliferate during glomerular development, differentiated podocytes have a quiescent phenotype[47]. This is required for podocytes to perform their specialized functions[47]. Two independent target prediction algorithms and luciferase assay identified p57[Kip2] as a relevant miR-92a target. The CDK inhibitor p57[Kip2] regulates cell proliferation and differentiation[48]. This protein is typically present in differentiated and post-mitotic non-renal cells. Several groups observed that the de novo expression of p57[Kip2] in podocytes during glomerulogenesis coincides with the acquisition of a terminally differentiated quiescent phenotype[6, 49]. Loss of p57[Kip2] expression in podocytes is recognized as an early feature of proliferative glomerular diseases and dedifferentiation in vitro[3, 6, 8, 49–52], although the mechanisms involved remained unclear.

Identifying p57[Kip2] as a key effector of mi-92a in podocyte drove us to test whether p57[Kip2] participates in the extrinsic stimuli-dependent abrogation of podocyte quiescence. Our results suggest that a reduction in p57[Kip2] protein, a brake on podocyte cell cycle, contributes to trigger activation of crescent formation in response to immune-mediated glomerular injury. We do not exclude that there are other potential miR-92a targets involved in the proliferative response of podocyte. For instance, the integrin subunit alpha5 is a target of miR-92a in ischemic tissues[20]. Alpha5 integrin subunit showed a gradual loss in early FSGS and became undetectable in advanced FSGS[53], meanwhile, its regulation in crescentic RPGN remain to be characterized. Change in integrin pattern may be important for podocyte functions in this setting, as shown in other conditions[54, 55].

The combination of our results from in vitro experiments, mouse models, and human tissues indicate that the high abundance of miR-92a can initiate a cascade of podocyte-destabilizing molecular events, starting with the downregulation of p57[Kip2] and proliferation. Mechanistically, binding of miR-92a to the target region of p57[Kip2] acts as negative regulator and causes a lack in the p57[Kip2] that is available for podocyte cell cycle resistance to ambient mitogenic stimuli. Moreover, specific blockade of miR-92a in vivo by an antagomir markedly prevented proteinuria, crescent formation, and renal failure. Together, these findings indicate that miR-92a control of podocyte phenotype may be a general paradigm for proliferative extracapillary diseases.

Several animal studies have used antagomirs to block target microRNA at a concentration ranging from 0.33 to 100 mg/kg of body weight. A potential concern of using high doses of these agents is that they may non-specifically block genes other than the target. Since previous studies have shown that a dose of 8 mg/kg produced an effective tissue response[20], we chose a similar, albeit slightly higher, dose for our own study. We felt that 12 mg/kg of body weight would account for the potential loss of drug in the urine of heavily nephrotic animals. Encouragingly, our results revealed a threefold decrease in miR-92 levels in glomeruli with this dose compared to the control antagomir. In contrast,

expression levels of 5 other microRNAs, miR-17, -18a, -19a, -19b, and -20a were not different between the two groups of mice, thus suggesting both efficacy and specificity of our antagomir. Interestingly, prolonged anti-92a blockade at higher dose for more than 10 weeks did not display detectable side effects in mice[45]. Another notable finding of our study is that STAT3 activation is required to trigger miR-92a pathogenic expression. This may lead one to consider anti-STAT3 strategies as anti-miR-92a options. Meanwhile, from a therapeutic perspective, the identification of a molecular target that is as cell-specific as possible may be important to limit the undesired side effects associated with more upstream ubiquitous targets such as STAT3.

Finally, we also provide proof of principle that delayed anti-miR-92a strategy could display therapeutic actions on glomerular function and structure in a severe model of RPGN. Although this treatment showed a preventive effect in our mouse model, it remains to be seen whether this holds true for human RPGN. Furthermore, although most of the pathophysiological actions of miR-92a were found in podocytes, it is important to note that anti-miR-92a strategies would be expected to alleviate endothelial inflammation, cardiac ischemia, and atherosclerosis[45], potentially important given the high risk of cardiovascular disease in individuals with RPGN[56–59].

## Methods

**Animals**. Mice with podocyte-specific GFP expression (*NPHS2*-Cre x mT/mG) were obtained by crossing podocin-Cre-positive mice[60] with mT/mG mice (Gt(2) 26Sortm4(ACTB-tdTomato,-EGFP)Luo/J)[61] on a C57BL6/J background that were purchased from The Jackson Laboratories (Bar Harbor, ME). Mice with podocyte-specific disruption of Stat3 were generated by crossing podocin-Cre-positive mice with *Stat3* floxed mice[62] on a C57BL6/J background. Age-matched littermates that had no deletion of Stat3 in any cells were considered as controls. To generate a time-specific and podocyte-specific knockout of miR-92a, we crossed mice carrying reverse tetracycline transactivator protein under control of the podocin promoter (iPod) with mice carrying the Tet-O-Cre transgene as previously described[13, 63] and with mice carrying a loxP-flanked miR-92a allele[64]. Doxycycline was administered for 3 weeks before administration of nephrotoxic serum after 1 week of washout. Age-matched littermates that had no deletion of miR-92a in any cells were considered as controls. Experiments were conducted according to the French veterinary guidelines and those formulated by the European Community for experimental animal use (L358-86/609EEC), and were approved by the Institut National de la Santé et de la Recherche Médicale and local University Research Ethics Committee (file 12–62, Comité d'Ethique en matière d'Expérimentation Animale, Paris Descartes).

**Induction of nephrotoxic nephritis**. Nephrotoxic nephritis was induced in male mice (10–12 weeks of age) by intravenous injection of 15 µl of sheep anti-glomerular basement membrane (GBM) nephrotoxic serum (NTS), which was diluted with 85 µl of sterile phosphate buffer solution as previously described[65, 66]. Serum injections were repeated twice: once on day 2 at 6 µl/g of body weight and a second time on day 3 at 7 µl/g of body weight.

**Biochemical measurements in blood and urine**. Urinary creatinine and blood urea nitrogen (BUN) concentrations were analyzed by a standard colorimetric method (Olympus AU400) at the Biochemistry Laboratory of Institut Claude Bernard (IFR2, Faculté de Médecine Paris Diderot). Urinary albumin excretion was measured using a specific ELISA assay for the quantitative determination of albumin in mouse urine (CellTrend GmbH).

**Human tissues**. Formalin-fixed, paraffin-embedded renal tissue specimens were obtained from the Hôpital Européen Georges Pompidou, Assistance Publique-Hôpitaux de Paris, Paris, France. Human tissue was used after informed consent by the patients and approval from, and following the guidelines of, the local Ethics Committee (IRB00003888, FWA00005831). Renal biopsy specimens with sufficient tissue for immunohistochemical evaluation after the completion of diagnostic workup were included.

**Histology**. Kidneys were harvested and fixed in 4% formol or in Alcohol-Formol-Acetic acid. Paraffin-embedded sections (5 µm thick) were stained by Masson's trichrome to evaluate kidney morphology and determine the proportion of crescentic glomeruli by a blind examination of at least 50 glomeruli per section. Silver staining of paraffin sections was also used for quantification of crescents.

**Immunohistochemistry and immunofluorescence**. Deparaffinized kidney sections were incubated for 30 min at 95 °C in target retrieval solution (S1699, Dako), then in peroxidase blocking reagent (S2001, Dako), were blocked in PBS containing 5% BSA, and were immunostained for phospho-STAT3 (Tyr705) (clone EP2147Y, Millipore, 1:50), nephrin (GP2, Progen, 1:400), WT1 (ab15249, Abcam, 1:400) STAT3 (D1A5, Cell Signaling Technology, 1:200), podocalyxin (AF1556, R&D systems, 1:100), and p57Kip2 (Clone M-20, Santa Cruz Technology, 1:200). For phospho-STAT3, and p57Kip2, staining was detected by Histofine reagents (Nichirei Biosciences), which contained anti-rabbit (414341F) or anti-goat (414161F) immune-peroxidase polymer for mouse tissue sections. WT1 and p-STAT3 primary antibodies were followed by a secondary rabbit anti-goat IgG AF488-conjugated antibody (Invitrogen, 1:400), p57Kip2 primary antibody was followed by a secondary anti-goat IgG AF594 (Invitrogen, 1:400), nephrin primary antibody was followed by a secondary anti-guinea pig IgG AF594-conjugated antibody (Invitrogen, 1:400). Podocyte culture cells were immunostained for podocin (ab50339, Abcam, 1:100), nephrin (ab58968, Abcam, 1:400), WT1 (ab15249, Abcam, 1:400), or p57Kip2 (Clone M-20, Santa Cruz Biotechnology, 1:200). The nuclei were stained using DAPI. Images were obtained with an Axioimager Z1 apotome (Zeiss) with Axiovision microscopy software. For quantification of p57-positive nuclei or p-STAT3-positive cells, 10 glomeruli per mice and five randomly chosen mice from each group were examined to calculate the number of stained nuclei.

**miR-92a in situ hybridization**. In situ hybridization was performed on 5 µm-thick kidney paraffin-embedded sections cut and fixed in PFA 4% for 10 min. Then sections were washed with 1× PBS and were acetylated for 10 min. After washes, sections were incubated with protein kinase K (Sigma-Aldrich) for 10 min at 37 °C. After subsequent washes, sections were incubated with hybridization buffer for 5 h at room temperature. miRNA probes (miR-92a probe double-DIG labeled LNA probes, Exiqon, final concentration 20 nM) were mixed with denaturation buffer and added to the sections and were incubated over night at 56 °C. U6snRNA probe (3′-DIG labeled LAN, probe, Exiqon) was used at 10 nM final concentration and as a positive control. The following morning, sections were washed in successively decreasing SSC buffers for 5 min at 56 °C (5× 1 time, 1× 2 times, 0.2× 3 times) and were then washed. Sections were incubated for 1 h in blocking solution (B1 solution + 3% fetal calf serum + 0.1% Tween-20), and were then incubated with anti-DIG AP antibody (Roche, 1:2000) over night at 4 °C. After washes, sections were incubated with NBT/BCIP (Promega) in NTMT + levamisole (0,2 mM/L) for 48 h in the dark at RT. NBT/BCIP was changed every 12 h. Slides were fixed in PFA 4% for 30 min and mounted with Fluoprep mounting medium (Biomerieux).

For fluorescent in situ hybridization (FISH), sections were incubated twice in freshly prepared 3% $H_2O_2$ for 3 min to inactive endogenous peroxidases after fixation, acetylation, and incubation with proteinase K. Slides were then rinsed three times in PBS. Then, sections were incubated with hybridization buffer for 1 h at 37 °C. MicroRNA probes (miR-92a, 5′-3′–Digoxigenin-labeled Locked Nucleic Acid probe, Exiqon, 100 nmol/L; U6snRNA, 3′-Digoxigenin-labeled Locked Nucleic Acid probe, 2 nmol/L) were mixed with denaturation buffer and then incubated with the sections over night at 56 °C. Washes with decreasing concentrations of SSC were the same as for ISH. Then, slides were again twice incubated in freshly prepared 3% $H_2O_2$ for 5 min and washed three times in PBS. Blocking solution (Tris + 3% fetal calf serum + 1% BSA) was applied to slide for 1 h at room temperature, then incubated in anti-DIG-FAB peroxidase (POD) (Roche) diluted 1:400 in blocking solution for 1 h at room temperature. After washes with PBS, TSA Plus Cy3 system working solution was applied onto the sections for 10 min at room temperature in the dark according to the manufacturer's protocol (PerkinElmer Life sciences). The slides were washed three times in PBS.

To assess whether miR-92a was localized specifically in glomerulus, sections were processed for double fluorescence staining to visualize the simultaneous localization of miR-92a (red; Cy3) and a primary antibody for WT1 (green; AF488), a podocyte-specific marker. Sections were incubated in demasking citrate buffer during 20 min at 95 °C and solution with a primary anti-WT1 antibody (Abcam) was applied on slide over night at 4 °C. The next day, slides were twice immersed in PBS and incubated with a secondary AlexaFluor488-conjugated antibody (Invitrogen) during 1 h at RT or with a Histofine-AEC system (DAKO and Nichirei Biosciences). Nuclei were stained with DAPI for FISH and sections were mounted using a drop of Fluorescent mounting medium (DAKO).

**Laser capture microdissection of glomeruli**. Laser microdissection was performed with a PALM® RoboSoftware 4.6 MicroBeam system (PALM Microlaser Technologies, Zeiss Micro Imaging, Munich, Germany) coupled to an inverted microscope Axio Observer.Z1. Serial 20 µm-thick cryosections, stored either at −80 °C for a maximum of 48 h or processed immediately, were spread onto polyethylene naphthalate (PEN) membrane-coated slides (Carl Zeiss Micro Imaging, Munich, Germany) previously treated for 20 min under UV exposure. After sections, the slide is stained with toluidine blue solution under RNase-free conditions. Glomerular tufts were delineated using a graphical computer wizard and isolated from surrounding tissue by laser catapulting into the cap of a single micro-centrifuge Eppendorf Tube® filled with 20 µl of RLT plus buffer/1% NucleoGuard reagent (Amsbio) for subsequent RNA isolation with the AllPrep DNA/RNA

Micro Kit (Qiagen). One cap per "section collection" was used and caps were replaced on their tube and stored on dry ice prior to RNA extraction.

**Transmission electron microscopy procedure**. Small pieces of renal cortex were fixed in 4% glutaraldehyde, postfixed in 1% osmium tetroxide, and embedded in epoxy resin. Ultrathin sections were counterstained with uranyl acetate and examined in a JEOL 1011 transmission electron microscope with Digital Micrograph software for acquisition.

**Glomeruli preparation and isolation of podocytes**. We used the magnetic bead method described by Takemoto et al. with appropriate modifications[67]. In brief, dynabeads perfusion was performed through the abdominal aorta and harvested kidneys were transferred in fresh Hank's buffered salt solution (HBSS). Then, kidneys were minced into 1-mm$^3$ pieces using a scalpel in digest solution (collagenase 210 U/ml (Gibco), DNase I 40 U/ml (Euromedex)) and incubated at 37 °C for 15 min on a rotator (100 rpm) The solution was pipetted up and down with a cut 1000 μl pipette tip every 5 min. After incubation, all steps were carried out at 4 ° C or on ice. The digested kidneys were gently pressed twice through a 100-μm cell strainer and the flow through was washed extensively with HBSS. After spinning down, the supernatant was discarded and the pellet resuspended in 2 ml HBSS. These tubes were inserted into a magnetic particle concentrator and the separated glomeruli were washed five times. Glomeruli were lysed in RIPA buffer for protein analysis or in Trizol for RNA study. For podocyte isolation, in a second digestion step, glomeruli were enzymatically and mechanically disrupted to yield a single cell suspension. Subsequently, green fluorescent protein (GFP)-positive podocytes from *NPHS2*-Cre x mT/mG animals were separated from the GFP-negative non-podocyte glomerular fraction by fluorescence-activated cell sorting (FACS) as published[68].

**Glomeruli isolation for culture of primary podocytes and in vitro assays**. Magnetic beads infused-mouse kidneys were extracted, minced, and digested in 2 mg/ml collagenase I solution (Gibco) in RPMI 1640 (Invitrogen) at 37 °C for 3 min, then filtered through a 70-μm cell strainer and once more through a 40-μm cell strainer. The homogenate was centrifuged at $720 \times g$ for 6 min and the cells were plated. Podocyte primary cultures consisted of freshly isolated glomeruli plated in 6-well dishes in RPMI 1640 (Invitrogen) supplemented with 10% fetal calf serum (Biowest) and 1% penicillin–streptomycin (Invitrogen). Purity of culture of differentiated primary podocytes was verified as previously described[13, 69] and shown in Supplementary Fig. 3b. Podocyte primary cultures used in this study was always P0. The outgrowth of podocytes started between days 2 and 3. Podocyte outgrowth area was quantified at day 4 using ImageJ software. Differentiated podocytes were exposed to HB-EGF (10 ng/ml, Preprotech), AG1478 (1 μM, Calbiochem), anti-mIL-6 monoclonal antibody MP5-20F3, monoclonal rat IgG1, κ isotype control immunoglobulin (both functional grade purified, 10 μg/ml, eBiosciences), Stattic (2 μM, Calbiochem), or recombinant IL6 (10 ng/ml, Preprotech) for 16 h. After stimulation, podocytes were scrapped in Phosphosafe buffer (Novagen) for protein extraction or in Trizol (Invitrogen) for total RNA extraction.

**miR-92a in vitro modulation**. MicroRNA-92a inhibition was achieved in vitro by transfecting primary culture podocytes with anti-miR-92a inhibitor using Hiperfect transfection reagent (Qiagen). Anti-miR-Control was used as a control (All from Ambion, 50 nM).

**p57$^{Kip2}$ 3′UTR luciferase assay**. For validation of p57 as a target of miR-92a, 3′UTR or mutated 3′UTR of mouse p57$^{Kip2}$ were cloned into a mammalian expression vector with dual luciferase reporter system (GeneCopoeia). HEK293 cells were transfected using Hiperfect (Qiagen). Transfections were performed using 1 μg dual luciferase reporter plasmids and a final concentration of 100 nM synthetic miR-92a mimic, or miR-126 as an irrelevant miRNA mimic (Applied Biosystems). Twenty-four hours after transfection, dual luciferase assays were performed using Luc-Pair miR luciferase assay kit (GeneCopoeia) according to the manufacturer's instructions. Firefly luciferase activity was normalized to Renilla luciferase expression control.

**p57$^{Kip2}$ silencing in vitro**. p57$^{Kip2}$ was silenced using ON-TARGETplus mouse p57 siRNA SMARTpool (Dharmacon). Primary podocytes were transfected with 50 nM ON-TARGETplus mouse p57 siRNA SMARTpool using Hiperfect following the manufacturer's instructions over night. Thereafter, medium was changed and cells were harvested after 72 h.

**Western blotting**. Proteins were extracted from glomeruli or podocytes with lysis buffer and were quantified by BCA protein assay (iNtRON Biotechnology). Samples were resolved on 4–12% Bis-Tris Criterion XT gels (Bio-Rad) then transferred to a polyvinylidene difluoride membrane. Membranes were incubated with the appropriate primary antibodies: rabbit anti-phospho-EGFR (Tyr1068) (D7A5, Cell Signaling Technology, 1:1000), rabbit anti-EGFR (D38B1, Cell Signaling Technology, 1:1000), rabbit anti-STAT3 (D1A5, Cell Signaling Technology, 1:1000), rabbit anti-phospho-STAT3 (Tyr705) (D3A7, Cell Signaling Technology, 1:1000),

goat anti-p57$^{Kip2}$ (Clone M-20, Santa Cruz Biotechnology, 1:500). Protein loading was monitored by the rat anti-tubulin antibody (ab6160, Abcam, 1:5000). Secondary antibodies were donkey anti-rabbit HRP and donkey anti-goat HRP (GE Healthcare Life Sciences) with no cross reaction to sheep serum. Antigens were detected by enhanced chemiluminescence (Supersignal West Pico, Pierce) using a LAS-4000 imaging system (Fuji). Densitometric analysis with ImageJ software was used for quantification. The uncropped versions of western blotting are shown in Supplementary Fig. 10.

**Real-time PCR**. Total RNA was extracted from mice glomeruli, cultured podocytes, or human biopsies with Trizol reagent according to the manufacturer's instructions (Invitrogen). Total RNA was reverse transcribed into cDNA using the Quantitect Reverse Transcription kit (Qiagen). cDNA and standards were amplified with the Maxima SYBR Green/Rox qPCR mix (Fermentas) using an ABI PRISM thermo cycler. The comparative method of relative quantification ($2^{-\Delta\Delta CT}$) was used to calculate the relative expression level of each target gene. Mouse or human GAPDH was used as an internal control. The data were presented as the fold change in gene expression. The following oligonucleotides served as primers: Mouse Ki67 forward 5′- CCTCAAAAGCAGACGAGCAAGA-3′, Mouse Ki67 reverse 5′- GAGAGTTTGCATGGCCTGTAGT-3′.

Following extraction by Trizol extraction, miRNA expression was determined using Taqman miRNA assay (Life Technologies) according to the manufacturer's protocols. U6snRNA was used as an endogenous control. Total miR content analysis of isolated glomeruli from control mice or NTS-challenged mice was performed with the miRCURY LNA$^{TM}$ Universal RT microRNA PCR Mouse&Rat panel I+II (Exiqon).

**In vivo miR-92a inhibition in mice**. For preventive strategy, antagomiR treatment (12 mg/kg) was started 3 days before NTS injection. AntagomiRs (VBC Biotech, Vienna) were delivered by retro-orbital IV injections under brief anesthesia. Second and third injections were performed on days 1 and 3 accompanying the NTS injection. For curative strategy, antagomir were injected on days 4, 5, and 8 after NTS. A scramble antagomiR (Antagomir-Control) was used as control. The sequences (AntagomiR-Control (anti-miR-ctrl): 5′-AAGGCAAGCUGACCCU-GAAGUU-3′ and antagomiR-92a (anti-miR-92a): 5′-CAGGCCGGGACAA-GUGCAAUA-3′) were obtained from a previously published study[20]. In the miR-92 antagomir and control antagomir, the 2′O RNA base are methylated and the first two bases and the last three bases are phosphorothiated to increase the stability of antagomir and hence protect it from degradation. In addition, a cholesterol-TEG was added at the 3′ for easy entry of the antagomir to the cells. AntagomiR-Control and antagomiR-92a were previously successfully used in vivo following administration in kidneys[42, 70]. Saline-treated mice were used as a control of the scramble antagomiR.

**Statistical analyses**. All values were expressed as means + s.e.m. When sample size was less than five per group, exact, two-sided comparisons were performed with exact test using StatXact 8.0 software (Cytel Software Corporation, Cambridge MA, USA). In other cases, the two-tailed Mann–Whitney test, was used as appropriate. For experiments with more than two subgroups, the nonparametric Kruskal–Wallis ANOVA followed by Dunn's multiple comparison test were used. Values of $p < 0.05$ were considered significant. Statistical analyses were calculated using Prism v5.04 software (GraphPad Inc, La Jolla, CA, USA).

**Data availability**. The authors declare that data supporting the findings of this study are available within the paper and its supplementary information files or from the corresponding author on reasonable request.

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

## Acknowledgments

This work was supported by INSERM, the European Research Council-ERC Grant 107037 to P.-L.T., and the Association des Malades d'un Syndrome Néphrotique (AMSN). We are grateful to La Fondation du Rein for supporting Dr G. Bollée and the French National Agency for Research (ANR Grant "SWITCHES" to P.-L.T.) for supporting Dr C. Hénique. T.B.H. was supported by the DFG (CRC1140, CRC 992, HU 1016/8-1), by the BMBF (01GM1518C), by the European Research Council-ERC Grant 616891, and by the H2020-IMI2 consortium BEAt-DKD. We also thank Elizabeth Huc and the ERI970 team for assistance in animal care and handling, Chantal Mandet for excellent technical assistance, Nicolas Sorhaindo for biochemical measurements (ICB-IFR2, laboratoire de Biochimie, Hôpital Bichat, Paris, France), and Alain Schmitt and Jean-Marc Masse for transmission electron microscopy (Institut Cochin, Paris, France). We thank Soraya Taleb for providing Stat3 lox/lox mice. We acknowledge administrative support from Véronique Oberweis, Annette De Rueda, Martine Autran, Bruno Pillard, and Philippe Coudol.

## Author contributions

C.H., G.B., and P.-L.T. designed research; C.H., G.B., X.L., N.D., F.G., M.C., L.G., H.L., C.M., I.B., and L.L. performed research; D.N., P.B., O.L., A.K., and E.T. participated in experimental design and provided tissues; C.H., G.B., X.L., and P.-L.T. analyzed data; C.H., N.D., G.B., O.L., A.T., A.K., E.T., P.B., T.B.H., L.M., and P.-L.T. wrote and/or discussed the paper; P.-L.T. participated in experimental design, supervised, and funded the project and wrote the manuscript. All authors discussed the results and commented on the manuscript.

## Additional information

**Competing interests:** The authors declare no competing financial interests.

