## [Peer Review File · Nature Communications]

PEER REVIEW FILE

Reviewers' Comments:

Reviewer #1 (Remarks to the Author):

The paper is vastly improved. It's novel and interesting and provides a framework for podocyte failure and explains the consequences leading to rapid progressive glomerulonephritis.

The discussion needs improvement: the proliferative response of podocytes goes through their complex integrin system. Integrin activation pathways in podocyte injury by the urokinase receptor (4 papers in Nature Medicine) are not even mentioned, yet may be paramount to the authors' findings. This is particularly important because it has been shown that increased miR-92 expression results in increased repression of integrin $\alpha 5$ mRNA, a known target of miR-92a interactions. uPAR and various integrins, including $\alpha(5)\beta(1)$, are known to modulate integrin-dependent cell adhesion and proliferation.

Reviewer #2 (Remarks to the Author):

This is manuscript previously submitted to NMED, and is now being reviewed for Nature Communications. Some issues raised by this reviewer were addressed. Other issues were specific for NMED and will not be discussed here.

There are no major weaknesses. However, the corresponding author does need to read this manuscript thoroughly to correct dozens of trivial issues, only some of which are listed below:

- 1) Figure 1 is extensively mislabeled. Figure 1b is inappropriately constructed.
- 2) Legend of figure 1 pages 26 and 27: Replace MM by MN for Membranous Nephropathy. In the text replace "right panel" by "lower panel". Replace (a) by (d) in figure F.
- 3) Figure 2h not mentioned in the paper.
- 4) Figure 5 panel f the vertical axis title is missing
- 5) Authors use primary podocyte isolates without mentioning specifically whether they were P0 or P1
- 6) Table 1 the authors should specify which patients have RPGN and which were used as controls. Whereas the reviewer is a physician and understands who is what, a PhD scientist reading this table would be completely lost!
- 7) Supplementary Figure 4f: Please reconstruct this figure. When you show relative comparisons, show black bars in one group and white bars in one group. This depiction is misleading, since the

actual lox control group expression should be lower than the WT control group expression.

Reviewer #3 (Remarks to the Author):

This revised manuscript has now addressed all my specific comments. They have performed additional new experiments

Response to comments for NCOMMS-17-10528A

We have carefully considered all of the very helpful comments from the Reviewers and we attach a full response.

Reviewer #1 (Remarks to the Author):

The paper is vastly improved. It's novel and interesting and provides a framework for podocyte failure and explains the consequences leading to rapid progressive glomerulonephritis.

We are pleased that reviewer 1 finds the manuscript improved with novel and interesting data and was satisfied with our reply on his/her original comments and concerns.

The discussion needs improvement: the proliferative response of podocytes goes through their complex integrin system. Integrin activation pathways in podocyte injury by the urokinase receptor (4 papers in Nature Medicine) are not even mentioned, yet may be paramount to the authors' findings. This is particularly important because it has been shown that increased miR-92 expression results in increased repression of integrin $\alpha 5$ mRNA, a known target of miR-92a interactions. uPAR and various integrins, including $\alpha(5)\beta(1)$, are known to modulate integrin-dependent cell adhesion and proliferation.

We thank the reviewer for his/her suggestion to improve discussion. We had omitted to mention the seminal studies demonstrating a role for the urokinase receptor in FSGS because 1/ its role has not been shown in crescentic RPGN and 2/ interaction of SuPAR with the $\alpha\beta 3$ integrin has been suggested, not with the $\alpha 5$ subunit. The integrin alpha 5 and integrin alpha V subunits are encoded by distinct genes. In fact, no direct interactions could be demonstrated between $\alpha(5)\beta(1)$ and either uPAR, uPA or uPA-uPAR complex (PMID: 19404550). Meanwhile, we acknowledge the relevant suggestion of the Reviewer and we have added in this paragraph, sentences about the miR-92a-mediated repression of integrin alpha 5 and its potential role in podocyte pathophysiology.

Reviewer #2 (Remarks to the Author):

This is manuscript previously submitted to NMED, and is now being reviewed for Nature Communications. Some issues raised by this reviewer were addressed. Other issues were specific for NMED and will not be discussed here.

There are no major weaknesses. However, the corresponding author does need to read this manuscript thoroughly to correct dozens of trivial issues, only some of which are listed below:

1) Figure 1 is extensively mislabeled. Figure 1b is inappropriately constructed.

We apologize also for this mistake and we have correct it.

2) Legend of figure 1 pages 26 and 27: Replace MM by MN for Membranous Nephropathy. In the text replace “right panel” by “lower panel”. Replace (a) by (d) in figure F.

We apologize for this typing error and these discrepancy between the text and figures have now been scrupulously crosschecked.

3) Figure 2h not mentioned in the paper.

As suggested, we have added a reference to Figure 2h in the text at page 9.

4) Figure 5 panel f the vertical axis title is missing

We thank the reviewer for this remark, we have adjusted the size of the graph to not hide the axis title.

5) Authors use primary podocyte isolates without mentioning specifically whether they were P0 or P1.

We have added a sentence in Methods to precise that all podocyte primary culture used in this study was P0.

6) Table 1 the authors should specify which patients have RPGN and which were used as controls. Whereas the reviewer is a physician and understands who is what, a PhD scientist reading this table would be completely lost!

We thank the referee for this remark and we have improved understanding of the table by adding a column with “groups of patients” specifying control or RPGN.

7) Supplementary Figure 4f: Please reconstruct this figure. When you show relative comparisons, show black bars in one group and white bars in one group. This depiction is misleading, since the actual lox control group expression should be lower than the WT control group expression.

We have taken note of this remark and corrected the figure.

Reviewer #3 (Remarks to the Author):

This revised manuscript has now addressed all my specific comments. They have performed additional new experiments

We thank the Reviewer for accepting our responses.